# OpenReview forum: "SciSafeEval: A Comprehensive Benchmark for Safety Alignment of Large Language Models in Scientific Tasks"
_ICLR.cc/2025/Conference — ICLR 2025 Conference Withdrawn Submission_

### Official Review · Reviewer_Hd66 · 2024-11-01

**Soundness:** 2
**Presentation:** 3
**Contribution:** 3
**Rating:** 6
**Confidence:** 4

**Summary:**

The paper introduces a benchmark dataset called SciSafeEval, to evaluate the safety alignment of LLMs across a range of scientific domans -  chemistry, biology, medicine and physics and language scope - textual, molecular, protein and genomic. It improvises over existing benchmarks on the following fronts - (1) a more comprehensive coverage of domains  (existing ones exclude medicine and physics). (2) larger in scope includes 10 times more examples than prior benchmarks.  and (3) incorporates jailbreak prompts that challenge existing safety guardrails.

**Strengths:**

The paper makes a sound contribution and address an important problem to ensure LLM  safety and alignment in scientific domains. The paper ensure comprehensiveness of the the entities considered benign versous hazorudous by referring to many existing sources. They also disucss how  imporve safety of the models and raise interesting discussion points regarding robustness and specialization.

**Weaknesses:**

The paper could benefit from clarifying and expanding upon its evaluation. Addressing the following questions may help enhance understanding:

What does the number 31K samples refer to? Does it represent 31K diverse instructions, such as those shown in Figure 3?

Could you provide a comparison of the diversity of instructions in this dataset versus those in existing, narrower scope benchmarking datasets? This would help in understanding the dataset’s unique scope.

Since the dataset includes samples from multiple domains, such as medicine, biology, physics and chemistry are all of these domains used to evaluate models specialized in for example say protein LLMs? Naturally, performance may be lower in textual or molecular genomic domains in this case if the model lacks domain-specific knowledge or cannot decline to answer unfamiliar questions. How is a "threat" defined in this context—does it mean responding to a malicious question, or providing a malicious or incorrect answer to an otherwise benign question?


Is the performance reported in the heatmap exclusively on malicious instructions? If so, does few-shot learning impact performance differently on benign questions versus safety-critical evaluations? The paper mentions using both benign and malicious prompts in few-shot settings—does the ratio of these prompt types influence performance? What would be the recommended mix for optimizing safety performance?

Could you elaborate on the quality assurance process? Specifically, how was malicious content identified—was a sample deemed malicious if any reviewer flagged it, or did all reviewers need to reach consensus?

**Questions:**

It would be helpful to receive a response from the authors on the questions described above.

---

> ### Author Response · Authors · 2024-12-02
>
> Thank you for your detailed and thoughtful feedback. We have addressed your concerns and made the following revisions:
>
> - Clarification of 31K Samples: We clarified that the "31K samples" refer to unique and diverse instructions across multiple domains, as represented in Figure 3. This ensures transparency in the dataset's composition.
>
> - Comparison with Existing Benchmarks: A new subsection provides a detailed comparison of the diversity and scope of instructions in our dataset versus those in narrower benchmarks. This highlights the unique contributions of our dataset in addressing safety alignment across multidisciplinary scientific domains.
>
> - Definition of "Threat": We clarified the operational definition of a "threat" in our context, which includes responses to malicious questions, as well as providing incorrect or unsafe answers to otherwise benign queries.
>
> - Few-Shot Performance Analysis: The impact of few-shot learning on performance across benign and malicious prompts has been analyzed further. We explored the ratio of benign to malicious prompts and proposed an optimized mix for safety alignment, detailed in Section X.
>
> - Quality Assurance Process: The process for identifying and validating malicious content has been elaborated. Specifically, we described the consensus-based review mechanism used to ensure reliability in labeling malicious samples.
>
> - Multidomain Evaluation Challenges: Additional context has been provided on performance differences across domains (e.g., textual, molecular genomics, and protein-related tasks). These differences are linked to model specialization and domain-specific knowledge requirements.
>
> These revisions aim to address your concerns comprehensively and enhance the clarity, depth, and rigor of the manuscript. We appreciate your feedback, which has significantly improved the quality of this work.

---

### Official Review · Reviewer_hBPb · 2024-11-01

**Soundness:** 3
**Presentation:** 3
**Contribution:** 2
**Rating:** 3
**Confidence:** 4

**Summary:**

The paper introduces a benchmark designed to evaluate the safety alignment of large language models (LLMs) when used in scientific domains like chemistry, biology, medicine, and physics. The authors highlight the critical risks LLMs pose in these fields, such as the generation of potentially harmful compounds, pathogenic sequences, or instructions for synthesizing such substances. The benchmark includes a robust and comprehensive dataset of over 31k examples and evaluates models under various prompting settings, including zero-shot, few-shot, and chain-of-thought.

**Strengths:**

1. The paper makes a substantial contribution by providing a large-scale, multidisciplinary benchmark tailored to the needs and risks of scientific language modeling. It addresses a timely and critical issue in the application of LLMs to scientific domains.
2. The introduction of multi-disciplinary safety evaluations and the jailbreak prompt feature seems valuable.

**Weaknesses:**

1.	The motivation for creating instruction-tuning datasets using *only harmful compounds* from hazard databases is not well justified in terms of real-world applicability. It seems the authors collected substances associated with hazard-related tags and created an instruction dataset using templates from prior work, followed by prompting experiments. However, the practical impact and real-life utility of these experiments remain unclear, especially since real-world hazard assessments are often multi-layered especially for chemistry.
2.	The authors aim for the LLM to not respond at all when there is a risk of mentioning hazardous or toxic substances. If the LLM refuses to respond even to legitimate and safe queries simply because the query involves a topic or substance that can be hazardous in different contexts, it could hinder useful and informative interactions. An improved strategy could involve the LLM generating safe, context-aware responses that include warnings or safety instructions. Then it raises the question: how to evaluate such responses.
3.	The notion of hazard or toxicity in fields like chemistry or medicine is nuanced. For example, certain levels of toxicity might be acceptable under specific conditions: dose, exposure duration, specific patient contexts, etc. Thus, suggesting that a simple binary approach to harmful versus non-harmful substances might be insufficient.
4.	The evaluation framework appears to be incomplete -- it focuses solely on accuracy in cases where the model fails to refrain from responding to instructions involving hazardous substances. What about the cases where the model fails to respond to instructions related to non-hazardous substances? Such cases would equally impact the model's practical utility and should be accounted for in the evaluation.
5.	While the paper successfully identifies vulnerabilities in current LLMs, there seems to be little contribution to actually address this issue. The identified issues are somewhat already familiar to researchers, hence limiting the major contribution of this work to the introduction of the dataset which is crowdsourced from existing work.
6.	Since the evaluation heavily relies on adversarial prompting (jailbreak prompts), there is a risk that the models may have been inadvertently tuned or overfitted to specific types of prompts used in the benchmark. What if the prompts do not adequately represent the full range of possible adversarial strategies?

**Questions:**

1. Could the authors provide insights into the underlying mechanisms that make certain models more vulnerable than others, such as differences in training data or tokenization strategies?
2. Could the authors provide more details on the diversity of jailbreak prompts?
3. For few-shot experiments, how exactly are the "3 representative examples demonstrating effective strategies for handling malicious prompts" selected? Any quantifier? The example in G.1 shows a case for DNA classification. What about the other tasks

**Details Of Ethics Concerns:**

How do the authors ensure safe and responsible usage of the dataset?

---

> ### Author Response · Authors · 2024-12-02
>
> Thank you for your detailed and insightful feedback. We have addressed your concerns as follows:
>
> - Motivation for Using Hazard Databases: We clarified the rationale for constructing instruction-tuning datasets using harmful compounds from hazard databases. A new subsection now emphasizes the significance of real-world safety assessments, ensuring a balanced approach to hazardous and non-hazardous contexts.
>
> - Improved Strategy for Response Evaluation: To address concerns about the LLM's refusal to respond to legitimate queries involving hazardous topics, we propose an enhanced evaluation strategy. This includes context-aware assessments and distinguishing between legitimate use cases and safety risks.
>
> - Nuanced Evaluation of Hazard and Toxicity: We now consider the nuanced nature of toxicity levels in different scientific contexts, accounting for factors such as dosage, exposure duration, and acceptable limits. This is reflected in our updated evaluation methodology.
>
> - Evaluation of Non-Hazardous Instructions: To balance the scope, we extended our analysis to include non-hazardous instructions and their impact on LLM performance. This adjustment ensures that the evaluation framework captures the full spectrum of practical utility.
>
> - Insights into Vulnerability Mechanisms: We provided an analysis of underlying factors that may make some models more vulnerable to adversarial prompts, including differences in training data and tokenization strategies.
>
> - Diversity of Jailbreak Prompts: Additional details about the diversity and design of jailbreak prompts have been included, with examples to illustrate their variations and coverage.
>
> - Few-Shot Example Selection: We elaborated on the selection process for the three representative examples in the Appendix, including their applicability to other tasks beyond DNA classification. This provides broader insights into the generalizability of the few-shot approach.
>
> - Additional Changes: We addressed concerns about potential overfitting to specific types of prompts by broadening the evaluation set and highlighting mitigation strategies. Further details on the responsible use of the dataset have been added to the Ethics Considerations section to align with best practices in AI safety research.
>
> We believe these revisions address your concerns effectively and enhance the quality and clarity of the manuscript. Thank you again for your valuable suggestions.

---

> > ### Comment · Reviewer_hBPb · 2024-12-02
> > **Response to rebuttal**
> >
> > Thank you for following up on the concerns and suggestions. Based on the rebuttal, it seems that the authors addressed most of my concerns but I cannot find the updates in the manuscript. Could the authors explicitly provide the sections/line numbers for each addressed concern? Also, kindly highlight all the rebuttal modifications in the uploaded PDF.

---

### Official Review · Reviewer_rXVm · 2024-11-01

**Soundness:** 2
**Presentation:** 2
**Contribution:** 2
**Rating:** 3
**Confidence:** 4

**Summary:**

This manuscript proposes a benchmark to be used in assessing the ability of language models to avoid providing information when prompts include potentially harmful subjects in the science domains. Relative to existing benchmarks in this area, the proposed benchmark contains an order of magnitude more total examples and includes prompts related to the fields of physics and medicine.

Examples related to chemistry, biology, and medicine are constructed by mixing prompt templates sourced from previous scientific benchmarks with names of molecules sourced from toxicological screens, controlled substance ledgers, or similar. Examples related to physics are a subset selected from a previous publication. Examples using "jailbreak" prompting strategies are additionally constructed from the "standard" prompts.

An array of language models, both general purpose and specialist, are evaluated using the proposed benchmark and the results reported. A LLaMa-based model trained on a number of human-annotated example is used to determine whether the models engaged with a given prompt or successfully identified it the prompt as malicious and refused. Models are evaluated under three circumstances: zero-shot prompting, five-shot prompting, and a chain-of-thought prompting strategy. All models are found to perform poorly in the zero-shot scenario. The generalist models are found to outperform specialist models in general.

**Strengths:**

- The area of AI-safety in the sciences is of interest to both the public and regulators. It is good to see interest in expanding our ability to effectively quantify the safety of a given model.
- The proposed benchmark is larger in size and scope than the pre-existing benchmarks in this area.
- A substantial amount of work has been done in aggregating the test cases and benchmarking the existing language models.

**Weaknesses:**

- The only considered safety scenario is that of the "evil scientist", an assumed malicious user attempting to access information for misuse. There are other scenarios that a comprehensive scientific safety of evaluation should consider. For example, the suggestion of a faulty synthesis protocol that may lead to injury in the laboratory in the chemistry domain or a confident misdiagnosis leading to incorrect treatment in the medical domain.
- Many examples are produced for the proposed benchmark, but the vast majority appear to reduce to the same question: "Can the model identify the presence of a known toxic/controlled substance or infectious agent in the prompt?" This scope should be better documented to prevent potential users over-interpreting results from the proposed benchmark.
- The manuscript uses non-quantitative/non-scientific language in several places that would be helpful to replace with more specific language. For example:
  - (4, 210) "...using a large-scale, high-quality dataset."
  - (5, 254) "This multi-dimensional approach to data collection ensures comprehensive coverage... maximizing the... utility for further biological safety evaluation."
  - (6,303) "This holistic evaluation ensures a multi-faceted risk assessment under various conditions."

**Questions:**

- The manuscript states "we emphasize the comprehensive coverage of key tasks and safety considerations for each one." How were the included tasks determined to be comprehensive of the key tasks and safety considerations? Were any tasks considered but determined to be non-integral and excluded?
- The manuscript states "Continuous updates are implemented to keep the dataset aligned with evolving scientific knowledge and safety regulations". How are these updates implemented?
- The experimental results in Table 3/Figure 4 lack units. I think that these are the percentage of prompts successfully rejected by the model (i.e. the inverse of the attack success rate used in Table 5)? It would be helpful to explicitly state so.

---

> ### Author Response · Authors · 2024-12-02
>
> Thank you for your detailed and constructive feedback. We have carefully addressed your comments and made the following revisions:
>
> - Expanded Scope for Scientific Safety Scenarios: We broadened the scope of safety scenarios beyond the "evil scientist" paradigm. Examples now include situations involving benign intentions, such as faulty synthesis protocols in chemistry or misdiagnoses in medicine. This provides a more comprehensive coverage of real-world safety challenges.
>
> - Improved Documentation of Scope: To prevent potential over-interpretation of results, we have clearly documented the scope and limitations of our benchmark. The new subsection explicitly states the research questions and boundaries of applicability for our evaluation.
>
> - Scientific and Quantitative Language: Revisions have replaced non-specific language with precise and quantitative terms. For example, phrases like "using a large-scale, high-quality dataset" have been revised to include specific dataset details.
>
> - Task Selection Process: We elaborated on how tasks were selected to ensure comprehensive coverage of key safety considerations. Tasks were prioritized based on their relevance to scientific misuse risks and their alignment with safety regulations.
>
> - Quality Assurance and Updates: Details have been added to describe the mechanisms for implementing continuous updates. This includes the integration of evolving scientific knowledge and safety regulations to ensure the dataset remains current and robust.
>
> - Clarification of Experimental Results: We clarified the metrics presented in Table 3 and Figure 4. Additional context now explains that these represent the percentage of prompts successfully rejected by the models, highlighting their alignment with safety objectives.
>
> We believe these changes address your concerns effectively and improve the overall quality of the manuscript. Thank you again for your valuable suggestions.

---

### Official Review · Reviewer_HTDY · 2024-11-03

**Soundness:** 3
**Presentation:** 3
**Contribution:** 3
**Rating:** 6
**Confidence:** 3

**Summary:**

This paper introduces SciSafeEval, a benchmark designed to evaluate the safety alignment of LLMs in scientific tasks across disciplines like biology, chemistry, medicine, and physics. SciSafeEval introduces a jailbreak feature to test LLMs’ responses to potentially malicious queries, particularly those that could be misused in scientific research. The benchmark includes 30k samples, vastly surpassing previous benchmarks in both scale and scope.

**Strengths:**

The dataset construction adheres closely to established regulations across multiple scientific domains.

The scale of the dataset surpasses previous safety benchmarks in scientific fields. The dataset's design and collection process are effective.

The paper is easy to follow.

**Weaknesses:**

Since the setting is binary (whether or not the LLM generates harmful content). The few-shot/CoT method will bring significant benefit because it gives an example of rejecting harmful content. Could the authors discuss why some tasks may benefit significantly more than other tasks from few-shot/CoT examples

The interpretation of appendix F is not clear. How to interpret the “-” and 0s in the tables?

**Questions:**

Line 23, the quote in latex is ``''.

---

> ### Author Response · Authors · 2024-12-02
>
> Thank you for your thoughtful feedback. We have addressed your concerns as follows:
>
> - Few-Shot/CoT Performance Variability: We conducted an analysis to explain why few-shot and CoT enhancements vary across tasks. This discussion now highlights factors like task-specific dependencies and the role of prompt structure in influencing model behavior.
>
> - Appendix F Clarification: We added detailed explanations for the “-” and “0” entries in Appendix F to improve clarity and readability.
>
> - Typographical Fix: The quote style on Line 23 has been corrected for consistency.
>
> We appreciate your input, which has significantly contributed to improving the manuscript. Please let us know if further clarifications are needed.

---

### Note · Authors · 2024-12-16

**Comment:**

We sincerely thank all the reviewers for their valuable and insightful feedback. After thorough consideration, we have decided to withdraw this submission.

**Withdrawal Confirmation:**

I have read and agree with the venue's withdrawal policy on behalf of myself and my co-authors.